# Area-time efficient pipelined number theoretic transform for CRYSTALS-Kyber

Ayesha Waris ⬤*, Arshad Aziz, Bilal Muhammad Khan ⬤

National University of Sciences and Technology (NUST), Pakistan

* ayesha.waris@pnec.nust.edu.pk

## Abstract

CRYSTALS-Kyber has been standardized by the National Institute of Standards and Technology (NIST) as a quantum-resistant algorithm in the post-quantum cryptography (PQC) competition. The bottleneck in performance of Kyber is the polynomial multiplication based on Number Theoretic transform (NTT). This work presents two parallel architectures adopting Multi-Path Delay Commutator (MDC) approach on target FPGA platform. Resource sharing technique is adopted to perform PWM operations using MDC NTT/INTT architecture. Moreover, we propose various optimizations at architectural level to minimize resource consumption such as FIFO-based memory units for buffering of input output coefficients, LUT-based modular multiplier and distributed-ROM memories for twiddle factor storage. The presented architectures are implemented on Xilinx Artix-7 XC7A100T-3 device using Vivado Design Suite 2022.2 and coded using Verilog HDL. Our BRAM and DSP-free designs achieve 68% improved area-time product with a comparable ATP for PWM operations. Additionally, the two-parallel MDC architecture outperforms state-of-the-art architectures, using 29% fewer resources.

## 1. Introduction

The security of current cryptographic algorithms is threatened by the emergence of quantum computers. The hard problems which form the basis of popular algorithms such as RSA [1] and Elliptic Curve Cryptography (ECC) [2] are at risk by using Shor's algorithm [3]. Therefore, to ensure the confidentiality of sensitive information, it is crucial to develop cryptosystems that can resist quantum computing [4] attacks. The National Institute of Standards and Technology (NIST) commenced a Post-Quantum Cryptography (PQC) [5] competition in 2016, to standardize quantum-resistant algorithms. After three rounds, CRYSTALS-Kyber [6] was announced as the winner in Public-Key Encryption category in 2022.

CRYSTALS-Kyber is based on Module-Learning with Error (M-LWE) [7] problem, which belongs to the class of Lattice-based cryptography (LBC) [8]. Also known as Kyber, the Key-Encapsulation Mechanism (KEM) provides three security levels

**Data availability statement:** All relevant data are within the paper and its Supporting Information files.

**Funding:** The author(s) received no specific funding for this work.

**Competing interests:** The authors have declared that no competing interests exist.

(Kyber-512, Kyber-768, Kyber-1024) comparable to the different levels of AES [9] (AES-128, AES-192, AES-256). The high-performance of Kyber is credited to the modular structure in MLWE which results in smaller and compact key sizes and fast computations.

Polynomial multiplication [10] is a pivotal operation in key-generation, encryption and decryption processes in Kyber. The computational power required for matrix-vector and vector-vector polynomial multiplication is optimized by employing Number Theoretic Transform (NTT) [11] algorithm for multiplication. NTT is a derivative of Fast Fourier Transform (FFT) algorithm and performs multiplication over finite field. The basic school-book multiplication requires $O(n^2)$ time complexity which is reduced to $O(nlogn)$ [12] in NTT. Therefore, an optimally designed NTT-based polynomial multiplication is fundamental for a high-performance Kyber.

Various software, hardware/software co-design and purely hardware implementations of NTT for CRYSTALS-Kyber have been proposed in literature [13–17]. The parallelization and reconfigurability offered by FPGAs make them an ideal and popular choice for executing computationally intensive operations. Providing an optimal balance between resource consumption, execution time, latency and power is a crucial challenge for embedded-system designers.

In this work, we have presented several hardware-level optimizations and implemented an accelerated and efficient Number Theoretic Transform (NTT) polynomial multiplier for CRYSTALS-Kyber on FPGA platform. The main contributions in this paper are as follows:

- Two parallel-pipelined NTT architectures are proposed based on multi-path delay commutator (MDC) approach.

- Configurable butterfly units are implemented, performing NTT, INTT and PWM operations.

- FIFO based memory units are designed for buffering input/output coefficients.

- Optimized data scheduling is presented for organizing the data input sequence before entering the FIFOs.

- Management and storage of twiddle factors is proposed using LUT-based distributed ROM.

- Modified multiplication-free Barrett reduction along with a LUT-based Karatsuba multiplier unit is implemented.

- Proposed DSP and BRAM-free NTT designs achieve 68% improved ATP for NTT/INTT and a comparable performance for PWM operations when compared with previous works.

The rest of the paper is organized as follows: literature review and preliminaries are given in section 2 and section 3, proposed NTT architectures are given in section 4, results comparison and analysis is given in section 5. Followed by conclusion and future work in subsequent sections.

## 2. Literature review

Hardware acceleration of CRYSTALS-Kyber relies on high-performance of polynomial multiplication. Various studies presented till date, adopt multiple approaches to implement number theoretic transform (NTT). These optimizations include designing an efficient memory access pattern, managing and storing twiddle factors effectively, and implementing a compact, resource-efficient modular multiplier. In literature, NTT for CRYSTALS-Kyber has been designed using either iterative or pipelined architectures, which are summarized below.

In [13], Xing et al have proposed an iterative design employing two unified radix-2 butterfly units (BUs). Each BU independently processes even, and odd input coefficients and are configured to perform NTT, INTT and PWM operations using short control codes. Bisheh et al. in [18] have presented NTT architectures utilizing one and two configurable butterfly units, performing NTT, INTT and PWM operations. Both designs achieve the same area-time product emphasizing the need of trade-off between execution cycles and area consumption. Three hardware architectures for iterative based NTT are proposed by Yaman et al in [14]. Lightweight, balanced and high-performance configuration utilizes one, four and sixteen BUs and each mode performs NTT, INTT and PWM operation.

Processing unit bi-core presented in [17], is configured in NTT, INTT and a binomial multiplier mode for PWM computation. Three iterative architectures are implemented on FPGA platform using one, two and four processing units. Ni et al. in [19] have presented a BRAM-free iterative architecture that performs NTT, INTT, PWM operations utilizing three small FIFOs. In [20], Guo et al. presented a conflict-free memory access and management scheme together with an iteratively computed NTT, INTT and PWM operations. In a subsequent study [21], mixed radix-2/4 approach is adopted by Guo et al. for designing an optimized unified NTT core. DSP-free, NTT based polynomial multiplier is proposed in [22], which uses look-up table based modular multiplier and exploits BRAM usage for memory management. Nguyen et al [23], have proposed BRAM-free configurable NTT architecture. The iterative design uses a reordering unit to organize the order of coefficients after every stage in the NTT/INTT computation. An iterative radix-4 based NTT/INTT core is presented by Sun et al. in [15] with a ping pong memory access method. An NTT/INTT core is implemented in [24], which extensively uses hardware resources for parallel computation. Nguyen et al. [25] have proposed a multi-stage pipelining NTT/INTT unit employing a modified K2-RED reduction unit.

A pipelined dual-path delay feedback (DDF) NTT architecture is proposed in [26]. Duong et al. [27] have proposed a mixed-radix, pipelined multi-path delay (MDF) architecture to implement NTT-based polynomial multiplication but consumes substantial hardware resources. NTT/INTT based on two-parallel multi-path delay commutator (MDC) NTT/INTT approach is proposed in [28] by Ni. et al, using different FIFOs for NTT and INTT data flow. The design uses a separate Karatsuba-based multiplier for PWM computation. In [29], MDC-based NTT/INTT core and Karatsuba-based PWM unit is implemented for Kyber and Dilithium, using the same FIFOs for NTT and INTT data flow.

## 3. Preliminaries

CRYSTALS-Kyber utilizes the polynomial ring structure $R_p = Z_q[x]/(x^n + 1)$, $n = 256$ is the degree of polynomial, where $q = 3329$ is prime modulus which fulfills $q \equiv 1 \pmod{n}$, $Z_q$ is the ring of integers modulo $q$ and $x^n + 1$ is the irreducible polynomial. Kyber employs (NTT) for polynomial multiplication in encryption, decryption and key generation processes. The NTT is a modular arithmetic adaptation of the Fast Fourier Transform (FFT) [30] that operates on integers in finite field. The $O(n\log n)$ complexity of NTT-based polynomial multiplication [12] is significantly lower than the $O(n^2)$ complexity of conventional schoolbook multiplication. Forward NTT operation transforms the input polynomials into NTT domain, where the coefficients are point-wise multiplied (PWM). Therefore, NTT operation can be represented by $c = INTT\{NTT(a) \circ NTT(b)\}$, where $\circ$ represents the PWM operation and $a$, $b$ are the input and $c$ is the output polynomial

Negative-wrapped convolution (NWC) [31], an optimized technique for polynomial convolution, is applied when $\psi$ exists, where $\psi$ is the 2n-th root of unity, and satisfies $\psi^2 \equiv w \bmod q$ *and* $q \equiv 1 \pmod{2n}$, with w is the primitive

n-th root of unity. NWC differs from conventional NTT, as positive and negative powers of $\psi$ are multiplied in forward and inverse process, known as post and pre-processing. For a 256-point input polynomial $a$ in Kyber, NWC based NTT-transformed output [32] vector is given by $A[i] = \sum_{j=0}^{255} a[i].w^{ij}.\psi^{-i} \bmod q$ where $0 \leq i < 255$. Scaling factor $n^{-1}$, negative powers of w and $\psi$ are multiplied with vector F to transform back NTT output into original domain and is given by $a[i] = n^{-1} \sum_{j=0}^{255} A[i].\psi^{-i}.w^{-ij} \bmod q$.

The parameters in Kyber are not NWC friendly, as condition $q \equiv 1\ (mod\ 2n)$ is not satisfied. The NTT can be calculated by either splitting the polynomial into even or odd parts or by using an incomplete FFT trick [33]. By employing the splitting approach, the degree of polynomial reduces to $n = 128$, which now satisfies the above stated condition. The incomplete FFT trick requires cropping the last stage of the NTT computation reducing the total number of stages in Kyber to seven. Truncated NTT makes PWM operation in Kyber different from the straightforward classical PWM and compute modular multiplication of two 1-degree polynomials. The operation consists of five multiplication and two addition operations and is given by [1] and [2].

$$H0 = A0B0 + A1B1w \tag{1}$$

$$H1 = A0B1 + A1B0 \tag{2}$$

Decimation in Time (DIT) and Decimation in Frequency (DIF) are two fundamental approaches to compute NTT. Both can be computed using Cooley-Tuckey (CS) and Gentlemen-Sande (GS) butterfly structure. Interested readers are referred to [34] for different configurations of NTT/INTT. CS and GS butterfly comprises of modular addition, modular subtraction and modular multiplication functions. The output only differs in the placement of the multiplier. Fig 1 (a) and (b) present the CT and GS butterfly structure with their respective inputs and outputs.

In the literature, NTT architectures have been implemented using iterative and pipelined architectures. Each of these design approaches offers distinct trade-offs in terms of area and performance. A simple NTT architecture consists of a butterfly unit and a memory unit, and the same hardware is reused to calculate intermediate outputs of each stage. This makes iterative designs area-efficient but results in higher processing time and increased latency. Conversely, pipelined architectures divide the NTT operation into multiple stages, depending upon the highest degree of polynomial $n$. Each stage has its own dedicated hardware, performing $n/2$ butterfly operations. Pipelined architectures process data in a continuous flow and employ FIFO-based reordering units for data permutations between the stages. Pipelined architectures achieve higher throughput and lower execution cycles at the expense of increase in hardware resources. The Multipath Delay Commutator is a popular pipelined architecture used for Fast Fourier Transform (FFT) computation [35]. It has been adopted to accelerate NTT implementations in several designs for Lattice-based cryptography algorithms. The architecture uses multiple paths to process data in parallel and employs a simple, BRAM-free control scheme.

## 4. Proposed architecture

NTT-based polynomial multiplication is the most complex and computationally intensive part of CRYSTALS-Kyber. To accelerate Kyber, an efficient and high-performance NTT multiplier is necessary. A BRAM-free, iterative NTT

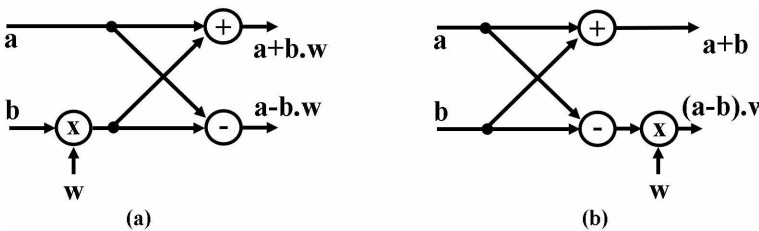

**Fig 1. (a) CT (b) GS butterfly.**

implementation is proposed for CRYTSTALS-Kyber in [23]. The NTT architecture is composed of a dual-butterfly that performs forward NTT, inverse NTT and point-wise multiplication (PWM) operation. MDC-based pipelined NTT architecture for CRYSTALS-Kyber is presented in paper [28]. The Radix-2 butterfly core in MDC chain performs NTT/INTT operation using resource sharing. Point-wise multiplication (PWM) is implemented using a separate hardware module.

In this study, two MDC-based architectures, a 2-parallel (MDC2NIP) and a 4-parallel (MDC4NIP) design, are presented along with elaborate discussion of execution of NTT, INTT, and PWM operations and data scheduling. Proposed MDC-based polynomial multiplication architecture configures the butterfly units to perform NTT, INTT, and PWM within a unified hardware unit, unlike the conventional MDC-based architectures. MDC2NIP and MDC4NIP outperforms in execution cycles when compared to NTT implementation in [23]. Integrating the PWM operation within the MDC chain significantly reduces resource consumption of presented architectures in comparison with paper [28].

Various optimization strategies are adopted in this work to achieve high performance, a few are mentioned here along with the motivation of their selection. In [29], Authors have proposed Radix-2 and Radix-4 MDC-based NTT architectures for CRYSTALS-Kyber. The butterfly units in the MDC chain are reconfigured to perform NTT and INTT operations. A separate hardware unit is designed for PWM operation. In the presented study, configurable butterfly units have been designed by employing resource sharing techniques. When cascaded, these units can perform NTT, INTT and PWM operations depending upon the controlled muxes. Adopting this technique helps save a considerable number of resources used in [29] for the PWM operation.

The [17], presents BRAM-based three iterative designs single butterfly unit, one dual butterfly unit and two dual butterfly unit. The architectures heavily use BRAM to store input/output and intermediate coefficients which increases the maximum clock frequency but results in high execution cycles and a larger resource utilization. The proposed designs MDC2NIP and MDC4NIP, utilize a pipelined architecture and a FIFO-based storage unit to implement NTT-based polynomial multiplication. This approach achieves computational efficiency and results in optimized execution cycles for NTT, INTT and PWM operations.

The FIFO-based memory stores input, output and intermediate values during polynomial multiplication operation. FIFO-based designs when compared to BRAM-based designs reduce memory access latency by continuously streaming data, enables higher throughput, achieves better pipeline efficiency, simplifies control logic and reduces the design complexity. The use of BRAM in this work is completely eliminated by using a LUT-based distributed BROM to store twiddle factors. This approach achieves computational efficiency and results in optimized execution cycles for NTT, INTT and PWM operations.

In [22], a binomial arithmetic core (Bi-core) comprising of four butterfly units is designed which reconfigures to perform NTT, INTT and PWM operation. The paper presents three different designs, comprising of 1 Bi-core (4 DSPs), 2 Bi-cores (8 DSPs) and 4 Bi-cores (16 DSPs). The high DSP usage, results in increased Equivalent number of slices (ENS) and ultimately in high Area-Time product.

Proposed, MDC2NIP and MDC4NIP incorporates 7 and 14 butterfly units respectively. Each butterfly unit has a modular adder, modular subtractor and modular multiplication unit. While modular adder and modular subtractors are straightforward, modular multiplier needs special consideration to design a resource efficient butterfly unit. Designing a DSP-based modular multiplier would significantly increase our equivalent number of slices (ENS) as MDC2NIP and MDC4NIP would require 7 and 14 DSP units. Hence, we have implemented a LUT-based Karatsuba multiplier integrated with an optimized Barrett reduction algorithm. This optimization has saved 63% of the equivalent number of slices compared to adopting a DSP-based multiplication approach.

In the following sections we have explained our architecture in bottom-up fashion. First, an optimized modular multiplication and configurable butterfly unit is presented. Followed by 2-parallel multi-path delay commutator (MDC2NIP) architecture with discussion of its three modes NTT (MDC2NTT), INTT (MDC2INTT) and PWM (MDC2PWM). Subsequently, the second proposed architecture, 4-parallel multi-path delay commutator (MDC4NIP) is presented with NTT (MDC4NTT), INTT (MDC4INTT) and PWM (MDC4PWM) configurations. The overall block diagram for 4-parallel architecture is presented in Fig 2.

## 4.1. Modular multiplication

While modular adder (MA) and modular subtractor (MS) are relatively simpler, implementing an efficient modular multiplication (MM) unit is complex. The choice of multiplier design depends on the specific algorithm being accelerated and determines the performance of the overall architecture.

Point multiplication is a fundamental operation in Montgomery curves, a special class of elliptic curves, used in Elliptic Curve Cryptography. The scheme involves computing multiples of a point on an elliptic curve expressed in the Montgomery form also known as point multiplication. A specialized hardware is designed consisting of modular multipliers to accelerate the operation. Montgomery algorithm is well suited for computations for point multiplication in Montgomery curves as the numbers are in Montgomery form. Interleaved multipliers perform multiplication and reduction at the same time and reduce the intermediate storage in FPGA based designs. Hence, integration of interleaved multipliers based on Montgomery multiplication in point multiplication operations optimizes computational efficiency. Using Lookup table-based multiplication method in point multiplication avoids full multiplications by storing precomputed values and thus reducing latency. The primary challenge is that large prime fields result in a larger memory overhead for storing precomputed results.

Montgomery and interleaved multipliers have been utilized in designing post-quantum digital signatures. Compared to Montgomery multiplication, which requires transformation into and out of Montgomery space and additional modular reductions, lookup table based multiplication eliminates these extra steps, leading to lower latency.

A soft-core multiplier is a configurable hardware multiplier implemented using lookup tables (LUTs), flip-flops (FFs) and shift registers instead of dedicated hardwired multipliers (DSP blocks). A soft-core multiplier consists of a partial product generator, adder tree and reduction logic. Lookup table multipliers are a type of soft-core multipliers that store the precomputed multiplication in LUTs. This multiplier architecture results in lower latency, fewer computational resources, less power consumption and faster modular multiplication when compared to Montgomery and interleaved multipliers.

In this work, we have designed soft-core modular multiplier by implementing Karatsuba-based integer multiplier followed by a Barrett-based reduction unit. The design details are given in the following sections.

**4.1.1. Integer multiplication.** Multiplication (MUL) unit performs integer multiplication of two 12-bit inputs $a$, $b$ and results in a 24-bit output, $c$. Taking advantage of parallelism in FPGAs and by adopting Karatsuba algorithm [36] we have broken down the 12x12 bit multiplication into two 6x6, one 7x7 multiplication, addition, subtraction and shifting operations. Karatsuba algorithm reduces the total number of multiplications from four to three and is given in eq 3 [13]:

$$a \times b = a_L b_L + ((a_L + a_H)(b_L + b_H) - (a_L b_L + a_H b_H))2^n + a_H b_H 2^{2n} \tag{3}$$

The hardware structure of MUL is shown in Fig 3. Input coefficients $a$ and $b$ are divided into 6 bit components $a_H$, $a_L$, $b_H$ and $b_L$. Sum of two six bit inputs results in a 7-bit output. Therefore, a 7x7 MUL, multiplies $(a_H + a_L)$ and $(b_H + b_L)$. Two 6x6 MUL units, perform $a_H \times b_H$ and $a_L \times b_L$. Before final addition of the intermediate outputs, multiplication by $2^n$ and $2^{2n}$ is performed by right shifting bits by a factor of 6 and 12, where $n = 6$.

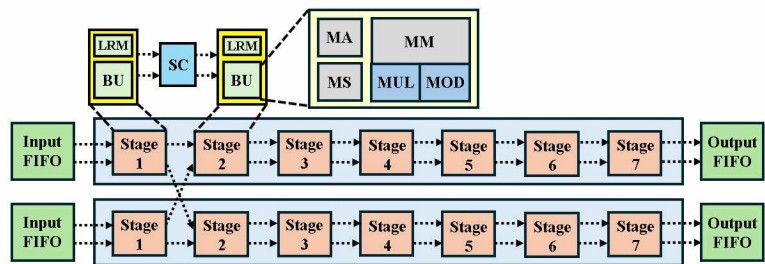

**Fig 2. Block diagram for MDC4NIP.**

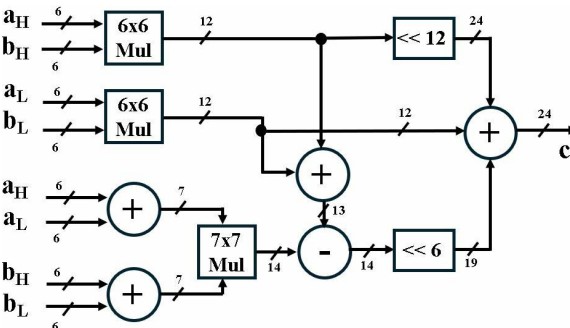

**Fig 3. 12x12 Multiplier unit.**

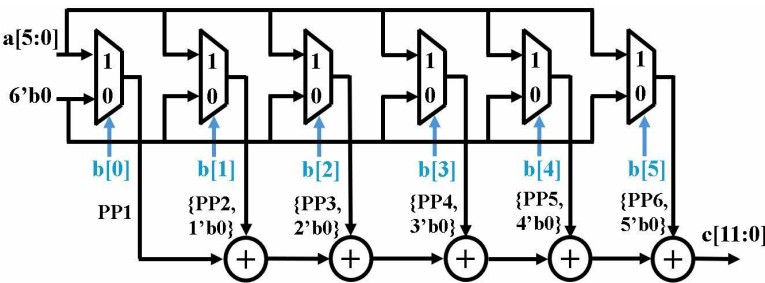

**Fig 4. 6x6 LUT-based multiplication.**

Fig 4 shows the proposed compact 6x6 multiplier. Implemented multiplier uses a set of muxes using the bits of multiplier, *b* as a selection signal. If selection bit is *1*, output of mux is multiplicand *a*, and if it is *0*, six bit *0*'s are directed to the output. Our design uses the approach of binary multiplication using AND gates where AND operation between multiplicand and a *0* or *1* bit results in *0* or multiplicand as output, respectively. The partial products are then concatenated with zeros to provide shift and are then added using five adders, like school-book multiplication.

#### 4.1.2. Modular reduction.

| Algorithm 1 Barrett Reduction Algorithm | |
|---|---|
| **Input:** *C* such that $0 \leq C < q^2$, $q = 3329$, precomputation $R = 2^{\log 2q}$, $\mu = R^2/q$ | |
| **Output:** $P \equiv C \bmod q$ such that $0 \leq P < q$ | |
| 1: | $C1 = \mu.(C/R)$ |
| 2: | $C2 = C1/R$ |
| 3: | $P = C - (C2.q)$ |
| 4: | **if** $(P \geq q)$ **then** |
| 5: | $P = P - q$ |
| 6: | **end if** |
| 7: | **return P** |

Barrette [37], Montgomery [38], and KRED [39] are the most frequently used approaches, with several modifications proposed over the years [16,40–42]. In this research we have presented an optimized Barrett reduction algorithm by replacing the constant multiplications in classical algorithm (Algorithm 1) by lightweight operations. Input *C* is the 24-bit product, *q* is the Kyber's modulus and *P* is the 12-bit reduced output. *R* and *μ* are the pre-computed values.

**Algorithm 2 Optimized Barrett Reduction Algorithm for CRYSTALS-Kyber**

**Input:** C such that $0 \leq C < q^2$, $q = 3329$, precomputation $\mu = R^2/q$, $R = 2^{log2q}$

**Output:** $P \equiv C \bmod q$ such that $0 \leq P < q$

| | |
|---|---|
| 1: | $C1 = C >> 12$ |
| 2: | $C2 = (C1 << 12) + (C1 << 10) - (C1 << 6) - (C1 << 4) - C1$ |
| 3: | $C3 = C2 >> 12$ |
| 4: | $C4 = (C3 << 12) - (C3 << 9) - (C3 << 8) + C3$ |
| 5: | $C5 = C - C4$ |
| 6: | $C6 = C5 - q$ |
| 7: | **if** ($C6 < 0$) **then** |
| 8: | $P = C5$ |
| 9: | **else** |
| 10: | $P = C6$ |
| 11: | **end if** |
| 12: | **return** $P$ |

We have applied the technique proposed in [43] and used it to implement our reduction unit (MOD). Algorithm 1 is optimized by replacing the constant multiplications in step 1 and step 3. The value of $\mu$ is calculated as 5039, which is a prime number and can be replaced by $2^{12} + 2^{10} - 2^6 - 2^4 - 1$. Moreover, modulus $q$ is also a prime number and can be substituted by $2^{12} - 2^9 - 2^8 + 1$. In hardware, multiplications by power of $2^n$ can be implemented as left bit shift by a factor of $n$. Therefore, multiplication by prime numbers is replaced by shift, addition and subtraction operations. $R$ is selected to be $2^{12} = 4096$ which equates to right 12-bit shifts in hardware. Final subtraction by $q$, ensure that the result satisfies $0 \leq P < q$. Implemented optimized reduction unit is presented in Fig 5.

### 4.2. Butterfly unit

Polynomial multiplier based on number theoretic transform comprises of NTT, PWM and INTT operation. Decimation in time (DIT) [44] configuration of NTT accepts inputs in normal order and generate output in bit-reversed order. Decimation in frequency (DIF), receives bit-reversed inputs and results in normal-ordered outputs. Implementing single DIT or DIF for NTT and INTT computation increases hardware requirements to execute extra bit-reversal operation. NTT and INTT consist of similar operations and are not performed simultaneously. Employing different DIT and DIF techniques in a single architecture results in a resource-inefficient design and under-utilization of resources.

In this research, bit-reversal operation has been averted by employing both GS and CT butterflies in a unified radix-2 butterfly unit. The NTT unit will accept data in normal order, resulting in a bit-reversed output which is then processed by PWM unit. Output of PWM unit is sent to INTT architecture which transforms back the result into normal order.

The butterfly unit presented in Fig 6, accepts three inputs $a, b, w$ and results in two outputs $O1, O2$. The BU consists of a modular adder (MA), a modular subtractor (MS) and a modular multiplier (MM). CT or GS butterfly is selected depending

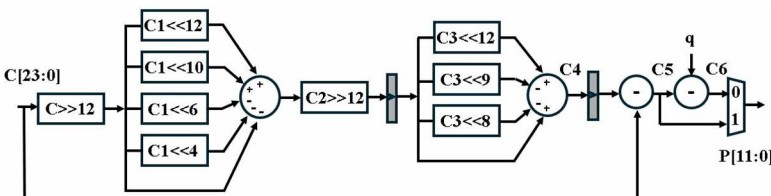

**Fig 5. Barrett reduction unit.**

on the control signal CT/GS given to the muxes. If CT/GS is *1*, butterfly is configured in CT mode. In case CT/GS is *0*, GS mode is selected.

We have adopted pipelining to reduce the critical path of architecture. The two pipeline stages added in the Barrett reduction unit (MOD) leads to inclusion of pipeline registers in the butterfly unit, to balance the outputs. In CT mode, pipeline registers are added in the input path of A, to synchronize the modular addition and subtraction of *a* with *b.w*. Pipeline stages are added in output path *O1* in GS mode to delay *a+b* and align it to the delayed output of modular multiplier *(a-b).w*. The proposed butterfly unit has a computational latency of two CCs whether working in CT or GS mode.

Final scaling is necessary at the output of GS butterfly by multiplying $n^{-1}$. Incorporating this multiplication causes additional resource utilization. Presented BU incorporates multiplication by $n^{-1}$, by dividing pre-computed twiddle factors for INTT operation by 2 and storing *w/2* in LUT- based distributed ROM, eliminating the need for extra multiplication, as done in [28]

Architecture in Fig 6 is modified by adding additional functionality to perform modular multiplication (MM) operation. Fig 7 presents the unified CT/GS/MM architecture, where added paths for MM computation are denoted by solid blue lines. CT, GS or MM operation is selected depending upon the control signal of muxes. Black muxes are operated by *CT/GS* signal while *NTT/PWM* controls the blue muxes. A low *NTT/PWM* signal, sets the *CT/GS* signal high by default and configures the BU in MM mode. When *NTT/PWM* signal goes high, the butterfly performs CT or GS operation, provided the *CT/GS* signal is high or low, respectively.

Fig 8 shows the BU in GS/CT mode, with non-active paths shaded grey. 8b presents architecture in MM mode. The output of modular multiplication operation is *a* at O1 and *(A.B)mod q* at O2, after a delay of two clock cycles. We have

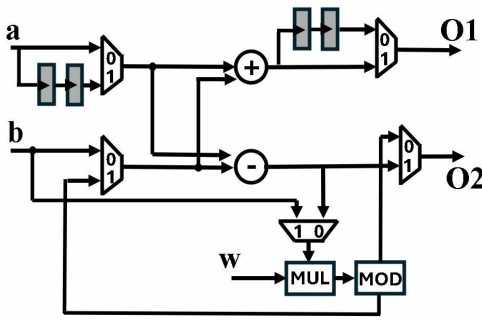

**Fig 6. Butterfly unit for CT/GS operation.**

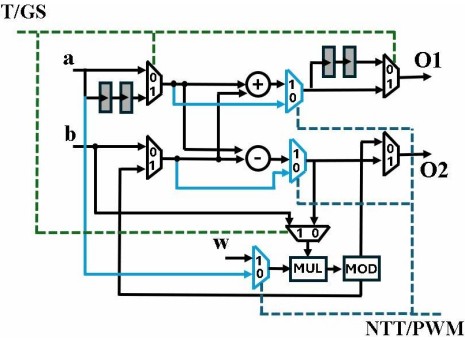

**Fig 7. Butterfly unit for CT/GS/MM operation.**

modified all the BUs in our designs to perform a specific operation for PWM computation. In this section we have only discussed modular multiplication, while the following sections discuss more details.

## 4.3. MDC2NIP

To compute polynomial multiplication using the Number Theoretic Transform (NTT), NTT operation is applied individually to each input and the transformed vectors are then point-wise multiplied (PWM). To finalize the multiplication and transform the result back to its original form, inverse NTT (INTT) is applied to the PWM result.

The proposed optimized MDC2NIP (Fig 9) parallel-pipelined design performs MDC-based NTT, INTT and PWM operations. Due to truncated property of Kyber explained in section 3, MDC2NIP has $log_2 n$-$1$ = $7$ stages, where $n$ = $256$. Each stage has one radix-2 butterfly unit, a set of muxes for data-path selection and FIFOs to provide offset for the selected mode.

Control signal *sel* given to the inter-BU muxes (IEM), configures the architecture to operate in NTT, INTT and PWM modes. If *sel* is *11,* NTT operation is performed (blue path); *10* configures the architecture in INTT mode (brown path); *01* selects PWM mode (green path) and *00* control code sets the architecture in the idle state. The input mux (purple) loads the coefficients from the memory and the data is continuously fed to the butterfly unit of the first stage. Similarly, the output mux loads the coefficients from the last stage into the memory. The muxes (black) selects the path for data flow for our specified operation. Subsequent sections discuss the data flow in all three modes of our architecture.

**4.3.1. MDC2NTT/INTT.** MDC2NIP configured in NTT mode for Kyber is shown in Fig 10 and consists of seven stages. The first six stages consist of a BU and a shuffling circuit (SC), and the last stage contains only a BU. Unified pipelined radix-2 BU in section 4.2 supports both CT mode for NTT and GS mode for INTT operation. The SC employs two FIFOs and a commutator to reorder data before it reaches the next stage. Each FIFO has a depth of $d$ = $n/(2.2^s)$, where $n$ = $128$ is the input sequence and $s$ ($s$ = $1,2,....,6$) is the number of the specific stage. The commutator consists of a set of muxes that swaps the input data depending on the control signal.

Data Scheduling for the shuffling circuit of first stage is illustrated in Fig 11(a) and 11(b). Since Kyber's modulus $q$ = $3329$ is not NTT friendly for $n$ = $256$, we split the input polynomial $P$ into two polynomials containing even $P_e$ and odd

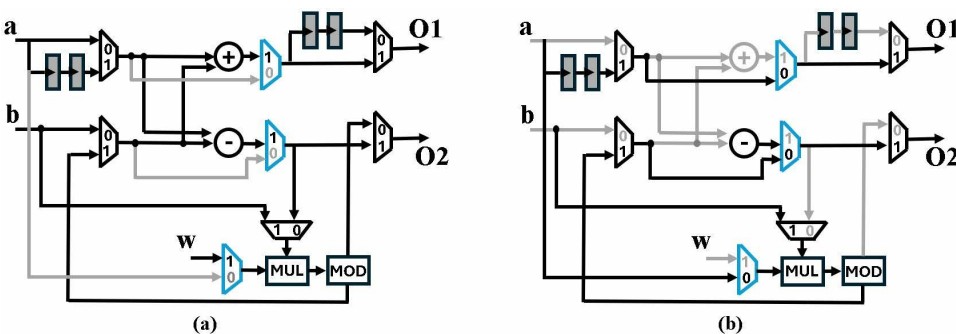

**Fig 8.** **(a) Active CT/GS paths and operators (b) Active MM paths and operators.**

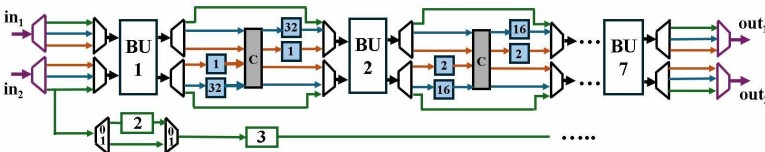

**Fig 9.** **NTT/INTT/PWM operations in MDC2NIP.**

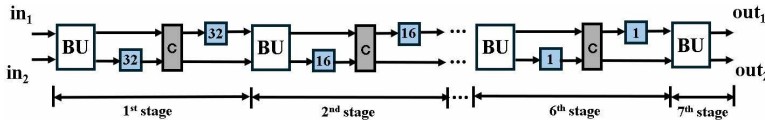

**Fig 10. NTT operations in MDC2NIP.**

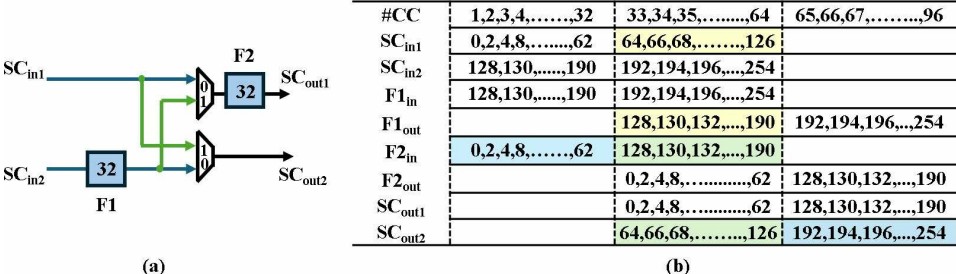

| #CC | 1,2,3,4,……,32 | 33,34,35,……...,64 | 65,66,67,……...,96 |
|---|---|---|---|
| $SC_{in1}$ | 0,2,4,8,…......,62 | 64,66,68,……,126 | |
| $SC_{in2}$ | 128,130,.....,190 | 192,194,196,...,254 | |
| $F1_{in}$ | 128,130,.....,190 | 192,194,196,...,254 | |
| $F1_{out}$ | | 128,130,132,...,190 | 192,194,196,..,254 |
| $F2_{in}$ | 0,2,4,8,……,62 | 128,130,132,...,190 | |
| $F2_{out}$ | | 0,2,4,8,…..........,62 | 128,130,132,...,190 |
| $SC_{out1}$ | | 0,2,4,8,…..........,62 | 128,130,132,...,190 |
| $SC_{out2}$ | | 64,66,68,……..,126 | 192,194,196,...,254 |

(a) (b)

**Fig 11. (a) Coefficient reordering in stage 1 NTT (b) Timing Diagram.**

coefficients $P_o$. This results in $n = 128$ and satisfies $q \equiv 1\ (mod\ 2n)$. The input coefficients are stored in four FIFOs (12x64), with each $P_e$ and $P_o$ having two FIFOs. Initially, $P_e$ is fed to inputs $SC_{in1}$ and $SC_{in2}$ in two parallel paths, with 1 coefficient arriving at each input per cycle. Depth of F1 and F2 is $d = n/(2.2^s) = 32$, where $n = 128$ and $s = 1$.

The control signal remains low for the first 32 clock cycles (cc), allowing input $SC_{in1}$ to pass through the mux while input $SC_{in2}$ is fed to the F1 to provide a delay of 32 cc. When the mux selection signal goes high, the last 32 coefficients of $SC_{in1}$ are swapped with the delayed first 32 coefficients of $SC_{in2}$. The shuffling is highlighted in yellow in Fig 11(b). F2 aligns the output $SC_{out1}$ with $SC_{out2}$ and the reordered data is sent to the second stage. In our design, the control signal for SC muxes is generated by counters, which have been designed independently in each stage.

The shuffling circuit (SC) provides the appropriate offset to the data for processing at subsequent stage. For the first stage of our Kyber MDCNTT, the required offset for input coefficients is 128. The coefficients are then permuted by the shuffling circuit and the (0,64) coefficient pair arrives at the second stage. The splitting, switching and rearranging of data in the first, second and last stage of NTT operation is shown in Fig 12, respectively.

MDC2NIP adopts pipelining to reduce the critical path and increase the maximum frequency of the overall architecture. The total delay can be calculated by summing the pipeline delay of the BU chain (7x2) and the delay incurred by the FIFOs (32 + 16 + 8 + 4 + 2 + 1) across the data path. The first NTT output coefficient will require 77cc after filling the internal pipeline stages and a complete NTT computation requires 77 + 128 = 205cc. Later NTT computations are performed in only 128cc.

Adopting resource sharing technique optimizes the architecture to perform both NTT and INTT operations using same BU and commutator. However, since the order of input coefficients in NTT and INTT are different, the FIFOs in the shuffling circuits cannot be reused. Fig 13 shows the data path and delay units in MDC2NIP architecture configured for INTT operation.

## 4.4. MDC2PWM

In Kyber, point-wise multiplication (eq 1 and 2) includes the modular multiplication between two 1-degree NTT-transformed vectors A ($A_0,A_1,A_2,... A_{255}$) and B ($B_0,B_1,B_2,... B_{255}$). The computation involves five multiplication and 2 addition operations.

Previous architectures in literature have designed a separate module for PWM and employed Karatsuba to reduce the number of multiplications. Proposed MDC2NIP performs PWM operation using the chain of butterfly units in MDC architecture. To simplify the calculation process, eq 1 and 2 are divided into smaller operations for computing $H_0$ and $H_1$ in Fig 14.

The reconfigurable BU in section 4.2 has a modular adder, modular subtractor and a modular multiplier unit. To design PWM architecture, we modified the seven BUs in MDC chain to perform modular multiplication/modular addition or butterfly operation depending on the selection signal for intra-module muxes (IAM). The signals of IAM generates the signal for IEM *sel* signal, as given in Table 1.

In PWM mode, coefficients ($A_i$ and $B_i$) are fed as inputs to the architecture on each cycle, where $i = 0,1,..,255$. First three BUs are dedicated to compute even coefficients of PWM output ($H_0$, $H_2$, ...., $H_{254}$). Configured as a modular multiplier, BU1 bypasses modular adder and subtractor operations. BU1 results in output S0 ($A_0B_0$) after a delay of 2 cc due to pipelined

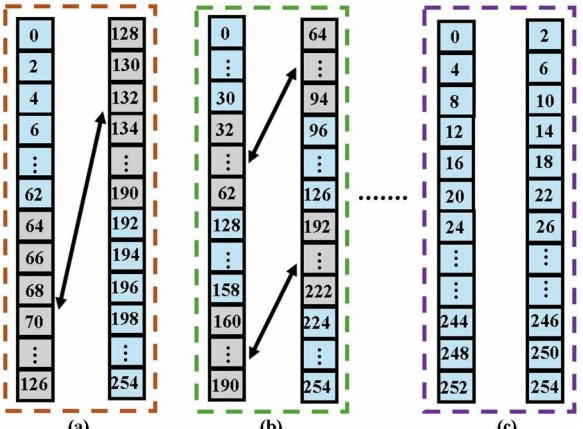

**Fig 12. Coefficient reordering in 7 stages of MDC2NTT (a) First stage (b) Second stage (c) Seventh stage.**

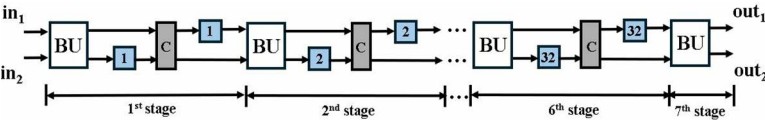

**Fig 13. INTT operations in MDC2NIP.**

| $H_0 = A_0B_0 + A_1B_1w$ | Module | $H_1 = A_0B_1 + A_1B_0$ | Module |
|---|---|---|---|
| $S_0 = A_0B_0$ | MM | $T_0 = A_0B_1$ | MM |
| $S_1 = A_1B_1$ | MM | $T_1 = A_1B_0$ | MM |
| $S_{1w} = A_1B_1.w$ | MM | $H_1 = T_0 + T_1$ | MA |
| $H_0 = S_{1w} + S_0$ | MA | | |

**Fig 14. PWM calculations broken into simpler operations.**

modular reduction unit, followed by S1, S2, …., $S_{255}$ per cc. BU2 configured in MM mode, multiplies S1, S3, …., S255 with 1 and S0, S2, …., $S_{254}$ with precomputed $w$, which is stored in distributed ROM memory. BU3 performs the modular addition (MA) and produces $H_0$, followed by $H_2$, $H_4$, …., $H_{254}$ on alternate cycles. The output is then delayed by 1 cycle using an external path, before being routed to the output *BU5out1*. Fig 15 presents the PWM operation in MDC2NIP.

The intermediate and final values of H0 are being generated and passed via the second path of the of the first three BUs, while the first path uses internal butterfly resources to delay and send coefficients $A_0$, $A_1$, $A_2$, …., $A_{255}$ to BU4. We have treated the pipeline registers BUs as storage units, which makes our design resource efficient.

BU4 and BU5 are dedicated for the computation of odd coefficients of PWM operation. The coefficients of vector B ($B_0$, $B_1$, …., $B_{255}$) are sent to BU4 through an external path and are rearranged using a reordering circuit (ROC). ROC uses two muxes and a FIFO of depth 2. Muxes in ROC goes low and high on alternate clock cycles, redirecting even coefficients ($B_0$, $B_2$, …., $B_{254}$) to FIFO and odd coefficients ($B_1$, $B_3$, …., $B_{255}$) to the output. The rearranged output coefficients $B_1$, $B_0$, …., $B_{254}$ are delayed synchronizing the arrival of inputs $A_i$ and $B_i$ to the *in1* and *in2* of BU4, respectively. BU4 configured in MM mode, generates the modular multiplication of $A_0B_1$, $A_1B_0$, …., $A_{255}B_{254}$ per cc. Modular addition of adjacent coefficients is performed by BU5 on every alternate cycle. Outputs $H_0$ and $H_1$ arrive at *BU5out1* and *BU5out2*, bypass BU6 and BU7 and are then loaded into the memory.

Fig 16 shows the internal architecture of connected BU1 and BU2 configured in MM mode. Blue path is the selection signal *NTT/PWM* and green path is for signal *CT/GS*. The control unit selects the mode based on the values *NTT/PWM* and *CT/GS* signals given in Table 1. Inactive operations and paths are highlighted in grey.

The timing diagram of proposed MDC based PWM is shown in Fig 17 presenting inputs and outputs of the first five BUs on each clock cycle. Even coefficients are completed before the computation of adjacent odd coefficients. Output $H_0$ arrives at *BU5out1* one cycle before $H_1$ at *BU5out2*. Each output port results in one coefficient on every alternate cycle. This implies that after filling the internal pipeline stages, PWM architecture loads one output coefficient in memory per cc. The pipeline delay across BU chain is *6* cc, and a complete PWM computation takes *6 + 256 = 262* cc. Subsequent complete PWM calculations are performed in 256 cc.

**4.4.1. Data scheduling.** In Kyber, each polynomial has 256 coefficients with a bit-width of 12-bits. Using even the smallest BRAM for data storage results in under-utilization of resources and parallel access of limited bit-width data

**Table 1. IAM and IEM muxes control code.**

| NTT/PWM (IAM) | CT/GS (IAM) | sel (IEM) | Mode |
|---|---|---|---|
| 1 | 1 | 11 | NTT(CT) |
| 1 | 0 | 10 | NTT(GS) |
| 0 | 1 | 01 | PWM |
| 0 | 0 | 00 | Idle |

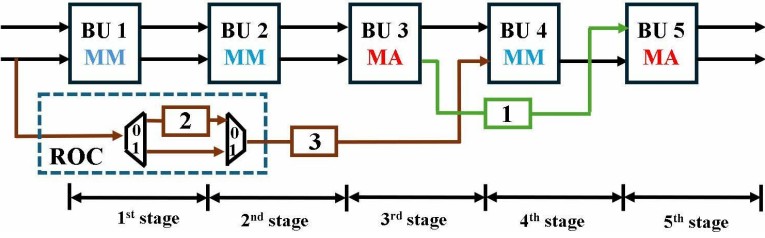

**Fig 15. PWM operation in MDC2NIP.**

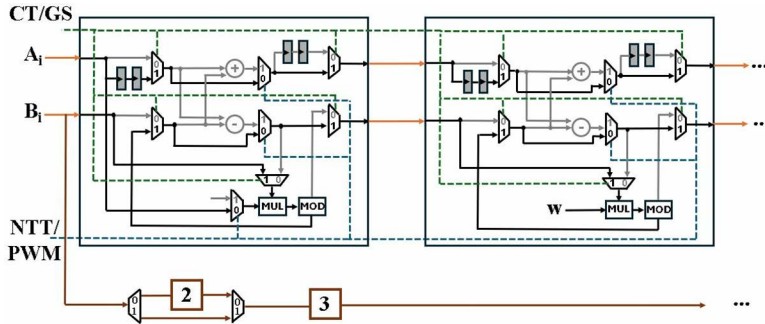

CT/GS

$A_i$

$B_i$

NTT/ PWM

**Fig 16. Module-Level diagram of first two stages in PWM operation.**

| CC | BU1 | | BU2 | | BU3 | | BU4 | | BU5 | |
|----|----|----|----|----|----|----|----|----|----|----|
| 1 | A0 | B0 | | | | | | | | |
| 2 | A1 | B1 | | | | | | | | |
| 3 | A0 | S0 | A0 | S0 | | | | | | |
| 4 | A1 | S1 | A1 | S1 | | | | | | |
| 5 | | | | | A0 | S0 | A0 | B1 | | |
| | | | A0 | S0 | A0 | - | | | | |
| 6 | | | | | A1 | S1w | A1 | B0 | | |
| | | | A1 | S1w | A1 | H0 | | | | |
| 7 | | | | | | | | | - | T0 |
| | | | | | | | - | T0 | H0 | - |
| 8 | | | | | | | | | - | T1 |
| | | | | | | | - | T1 | - | H1 |

**Fig 17. Timing diagram for PWM operation.**

components from memory results in reduced performance. In our MDC2NIP architecture, we have designed custom-made FIFOs for data storage and no BRAM unit is used which increases the hardware efficiency. Data elements from FIFOs can only be accessed in the order in which they are pushed into the FIFO. Hence, for NTT, INTT and PWM computation the order of input is altered before sending to the FIFOs.

For memory management and data accessing, 256 coefficients of a polynomial in Kyber are divided into four parts $P_{e1}$, $P_{e2}$, $P_{o1}$, $P_{o2}$ stored in FIFO Fa, Fb, Fc and Fd respectively. These four FIFOs each storing 64 data elements of 12 bits are collectively called Fm and is the main storage module of our architecture. For NTT operation, the data is arranged in FIFOs in a manner, that parallel inputs ($P_{e1}$, $P_{e2}$ and $P_{o1}$, $P_{o2}$) have an offset of 128. NTT module takes 2 coefficients as input and results in 2 output coefficients per cycle.

Even coefficients $P_{e1}$, $P_{e2}$ are processed first, followed by the computation of odd part $P_{o1}$, $P_{o2}$. In Fig 18, the black arrows indicate the FIFOs that send/receive data first while the blue arrows send/receive afterwards. As Fm will be unoccupied after data is fed to the NTT unit it will be reused to store the NTT output coefficients, optimizing memory management and reducing the storage resources.

NTT-transformed output coefficients in Fm are stored such that the coefficients in lowest part are ordered as *0,2,1,3*. This sequence of data facilitates PWM computation. Fv consists of coefficients from polynomial of matrix A or vector t, obtained during the key generation operation in Kyber. In PWM mode, each Fm and Fv provide one coefficient per cycle

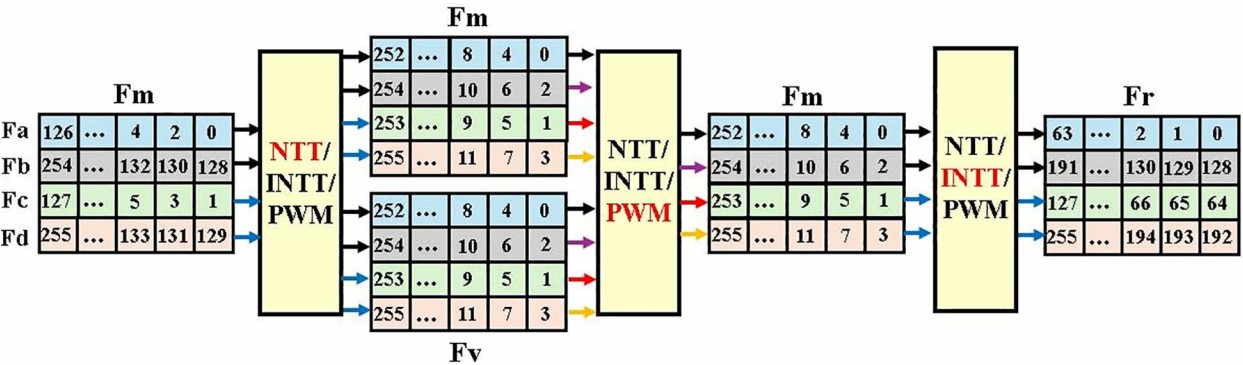

**Fig 18. Coefficient access scheme in MDC2NIP.**

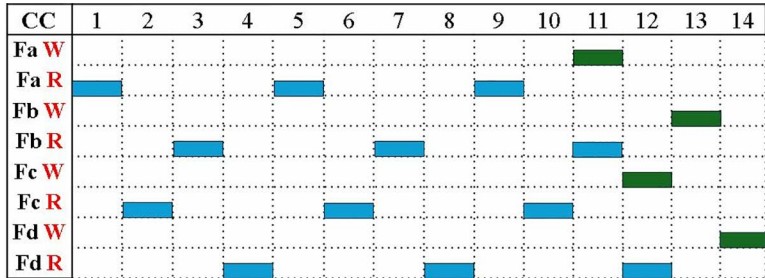

**Fig 19. Read Write operations in memory-based FIFOs.**

to the two parallel input and complete computation takes 256 cycles after filling the 6 internal pipeline stages. Output of PWM is again stored in Fm.

Write and read enable signals of FIFOs Fa, Fb, Fc and Fd, determine the data flow in and out of the PWM unit. Coefficient *0* is read from Fa, into the module on the first clock cycle. Fc, Fb and Fd read coefficients *1, 2* and *3* on subsequent clock cycles. This pattern continues for the remaining 256 cycles as shown in Fig 19. After filing the internal pipeline stages, the first PWM output coefficient *0* is stored back in Fa, which is now vacant. Similarly, coefficients *1, 2* and *3* are stored back in Fc, Fb and Fd. The PWM module generates one coefficient per cycle. Our proposed memory management approach organizes and separates the data in even and odd positions in memory based FIFOs before the INTT operation.

After PWM operation, MDC2NIP is configured to perform INTT. The output of PWM, is now fed to the INTT module. Coefficients of Fa and Fb are sent to the input for the first 128 cycles, followed by the coefficients of Fc and Fd. The parallel input coefficients have an offset of 2. The delay of the INTT module is the same as the NTT module. Result of a single NTT-based polynomial multiplication is stored in Fr, meanwhile Fm loads another set of coefficients to be transformed into NTT domain.

### 4.5. MDC4NIP

Exploiting the inherent parallelism in NTT, we have presented MDC4NIP architecture. The design has four parallel paths to process input polynomials, which reduces the computation cycle time by half compared to the previous proposed design. MDC4NIP consists of two MDC2NIP architectures operating in parallel, employing 14 BU cores and seven stages. Two unified radix-2 BU operate in one stage and simultaneously pass the data to subsequent stages in parallel.

The inter-BU (IEM) and intra-BU (IAM) muxes control the data path in Fig 20. The architecture can be configured in NTT (blue path), INTT (brown path) and PWM modes (green path). All stages have two LUT-based ROM memory associated with each butterfly unit to store twiddle factors for the three modes. BUs in MDC chain 1 and 2, perform operations independently and concurrently.

**4.5.1. MDC4NTT/INTT.** In NTT mode (Fig 21), all BUs are configured as a CT butterfly where each BU performs $n/4 = 64$ butterfly operations. Only two BUs are utilized in the first stage, whereas a shuffling circuit (SC) is excluded as data reorganization at different clock cycles is not required. Each of the next six stages incorporates two BUs and two SCs. The SC in the last stage organizes NTT output data to be executed by the PWM mode. Depth of FIFO in each stage can be determined by $d = n/(2.2^s)$, where $n = 256$ is the input sequence and $s$ ($s = 2,….,7$) is the number of the specific stage.

NTT module takes 4 coefficients from the memory and results in four output coefficients per cycle after filling the internal pipeline stages. The delay of the first complete NTT computation is 141cc, which is the sum of the pipeline delay of one BU chain (77cc) and the number of inputs in one path, ($n/4 = 64$). Subsequent NTT operations will only take 64 cycles.

The timing diagram of data reorganization in the first two stages of MDC4NTT is shown in Fig 22. FIFOs are represented vertically, where the top element is the first to exit. The 256 coefficients in a single polynomial in Kyber are divided into four sets, each having 64 coefficients. The data elements are organized in input FIFOs in a manner that the coefficient pairs (0,128) and (64,192) arriving at first stage BUs, are spaced at an offset of 128. Before exiting the first stage, coefficients of path 2 and path 3 are swapped in (a) to provide a 64 offset for the second stage. In (b), the two SCs operate to realign and synchronize data on clock cycles, providing an offset of 32 before it is fed to the third stage. (c) represents the transformed output coefficients at the end of NTT operation.

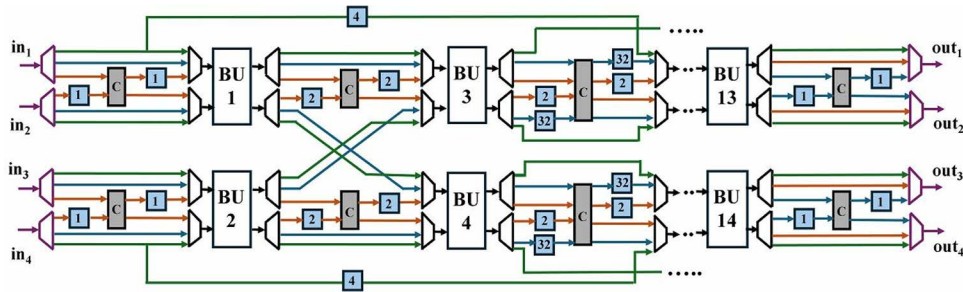

**Fig 20. NTT/INTT/PWM operations in MDC4NIP.**

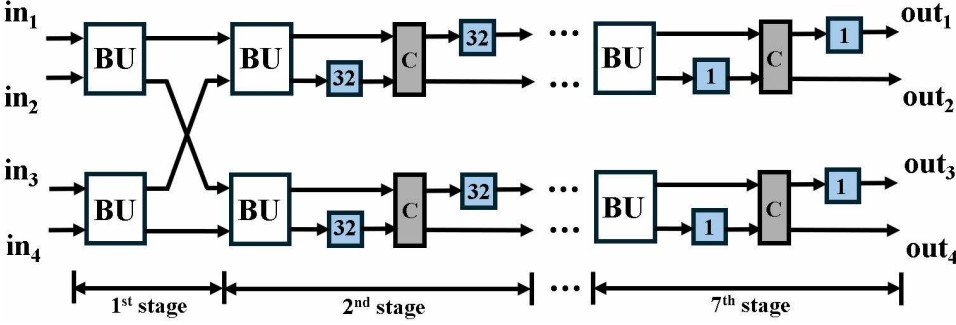

**Fig 21. NTT operations in MDC4NIP.**

                                                                 

Inter and intra-BU muxes select GS mode, when MDC4NIP is to be set in INTT mode. The two SCs in the first stage provide the correct offset of 2, for the operation. The first six stages consist of two SCs and two BUs, while the last stage only consists of two BUs. As NTT and INTT are symmetric, SC is not required in last stage. The depths of FIFOs are also reversed in inverse NTT operation as compared to forward NTT. INTT mode is presented in Fig 23.

**4.5.2. MDC4PWM.** Fig 24 shows MDC4NIP configured in PWM mode, which employs four stages to compute point-wise multiplication between two NTT-transformed vectors A and B. The calculation is broken into simpler operations in Fig 24. The BUs have a unified CT/GS butterfly structure which can also perform a specific operation to perform PWM calculation. Four inputs are fetched from the memory to the first stage BUs, per cycle. Configured in modular multiplication (MM) mode, BU1 computes product of even coefficients $(S0, S2,\ldots,S_{254})$ and BU2 results in product of odd coefficients $(S1, S3,\ldots, S_{255})$. Using advantage of the design of MDC4NIP architecture, coefficients $B_0, B_2,\ldots,B_{254}$ are swapped with $B_1$, $B_3,\ldots,B_{255}$, on each clock cycle, sending odd coefficients of vector b to BU3 and even to BU4. In the second stage, BU3 performs modular multiplication of $B_1, B_3,\ldots,B_{255}$ with *1* whereas, BU4 performs modular multiplication S1, S3,…,S255 with twiddle factor w. Outputs S0, S2,…,S254 and S1, S3 ,…,$S_{255}$ are sent to BU4 delayed by two units, where modular addition is performed resulting in $H_0, H_2,\ldots,H_{254}$ per cycle.

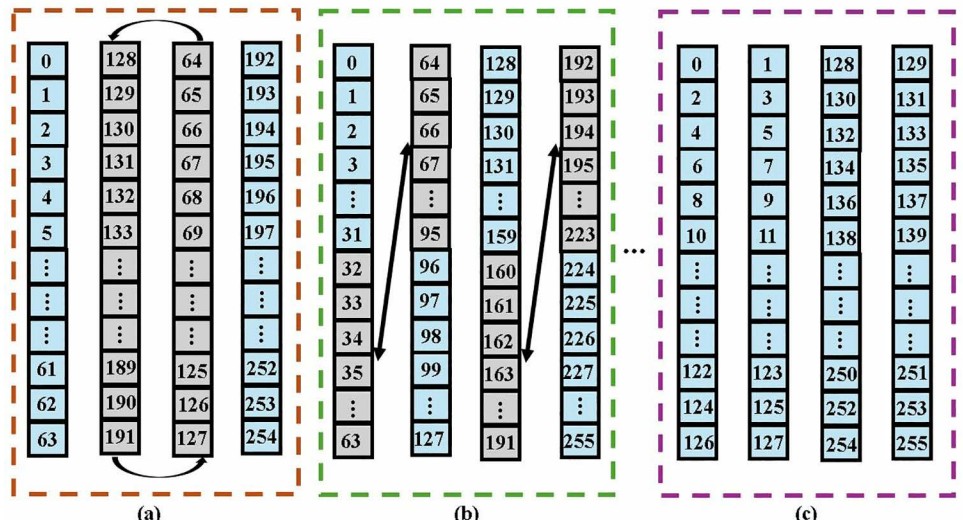

**Fig 22. Coefficient reordering in 7 stages of MDC4NTT (a) First stage (b) Second stage (c) Seventh stage.**

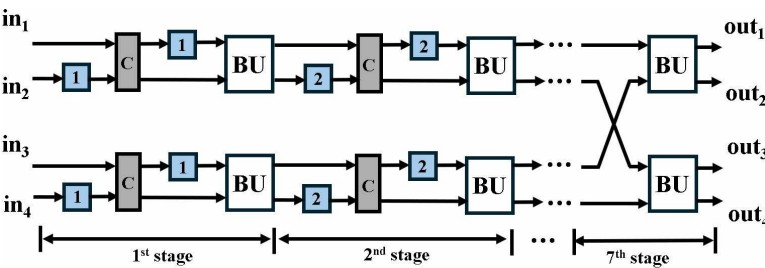

**Fig 23. INTT operations in MDC4NIP.**

To compute odd PWM outputs ($H_1$, $H_3$,…,$H_{255}$), alternate data path is provided to even and odd coefficients of A. The data elements are delayed by FIFO of depth 4, corresponding to the delay caused by BU1 and BU2. Coefficients $A_0$, $A_2$,…,$A_{254}$ and $B_1$, $B_3$,…,$B_{255}$ arrive at BU5 and coefficients $A_1$, $A_3$,…,$A_{255}$ and $B_0$, $B_2$,…,$B_{254}$, simultaneously. After a computational delay of 2cc, the output $T_0$, $T_2$,…,$T_{254}$ and $T_1$, $T_3$,…,$T_{255}$ are routed to BU8 for modular addition. The PWM coefficients ($H_0$, $H_1$,…,H255) are sent to the BU7 output ports passing through 5th, 6th and 7th stages, which remains inactive during the operation.

Fig 25 shows the internal architecture of the first two stages of MDC4NIP. Grey paths determine the inactive data routes and the selection lines for *CT/GS* and *NTT/PWM* mux are shown by green and blue paths, respectively. Even coefficients $A_i$ and $B_i$ are fed to chain 1 and odd coefficients $A_{i+1}$ and $B_{i+1}$ are given to chain 2, where i = 0,2,4,6,…,254. An alternate data path with FIFOs is provided to $A_i$ and $A_{i+1}$, for computation of odd coefficients of PWM operation.

The timing diagram for PWM scheduling operation in MDC4NIP is shown in Fig 26. Input and output data of BUs are highlighted blue and black, respectively. Four input coefficients are fetched from the memory and after filling the internal pipeline stages, two output coefficients are loaded into the memory per cycle. The total delay for the first NTT computation

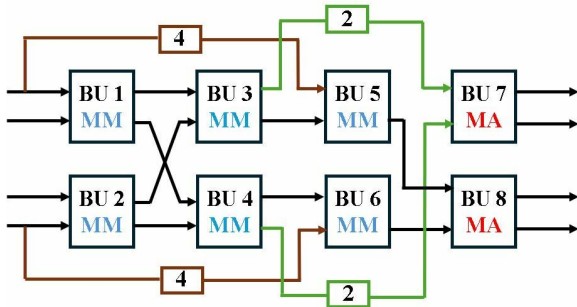

**Fig 24. PWM operations in MDC4NIP.**

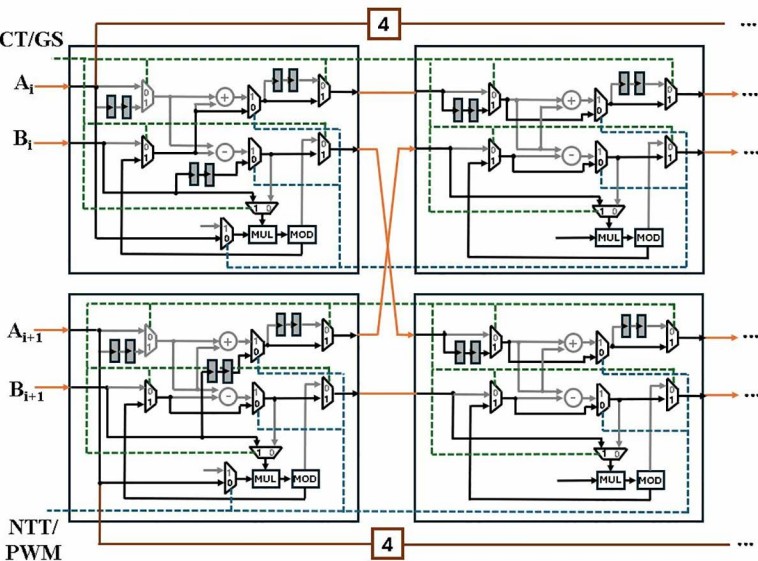

**Fig 25. Module-level diagram of first two stages in PWM operation.**

is $(n/2)+6 = 134$ cc, where $n = 256$ and 6 is the pipeline delay of the architecture. The following PWM computations will require 128 cc.

### 4.5.3. Data scheduling.

Data management in MDC4NIP is different than MDC2NIP as the increased number of BUs alters the memory access pattern. The design proposed utilizes FIFOs as the main memory unit for storing input and output coefficients. For NTT operation, data is divided into four sets $P_1$, $P_2$, $P_3$ and $P_4$ and stored in FIFO Fa, Fb, Fc and Fd, collectively called as Fm, as shown in Fig 27. Data elements in P1, P2 and P3, P4, spaced at 128 offset, are fed to BU1 and BU2 respectively. The NTT module takes 4 coefficients as input per cc and results in 4 output coefficients. After NTT operation Fm will be vacant and can be utilized for storing output NTT coefficients, increasing hardware efficiency. The output of NTT operation in Fm serves as input for PWM mode.

MDC4NIP takes 2 inputs each from Fm and Fv per cc when configured in PWM mode. The design results in 2 output coefficients per cycle after filling 6 pipeline stages. The output of PWM is stored again in Fm to enhance reusability of resources. The read and write enable signals determine the dataflow into and out of Fa, Fb, Fc and Fd during PWM operation. For the first 128 cycles the read signal of Fa and Fb is high, and data is fetched from the FIFO and given to the first stage of BU. After pipeline delay of 6 ccs, the write signal for Fa and Fb becomes active and the resulting two output coefficients are stored in Fa and Fb, which are now unoccupied. The write signal remains high for the next 128 cycles. Read signal of Fc and Fd becomes high at 129th cycle and the data coefficients are now routed from Fc, Fd to the PWM unit.

The output of PWM is organized in Fm in an appropriate sequence for inverse operation. In INTT mode, the input/output sequence is reversed compared to NTT. Output data will be stored in Fr, meanwhile Fm load data for next NTT computation.

| CC | BU1 | | BU2 | | BU3 | | BU4 | | BU5 | | BU6 | | BU7 | | BU8 | |
|----|----|----|----|----|----|----|----|----|----|----|----|----|----|----|----|----|
| 1 | A0 | B0 | A1 | B1 | | | | | | | | | | | | |
| 2 | | | | | | | | | | | | | | | | |
| 3 | S0 | B0 | B1 | S1 | S0 | B1 | B0 | S1 | | | | | | | | |
| 4 | | | | | | | | | | | | | | | | |
| 5 | | | | | S0 | B1 | B0 | S1w | A0 | B1 | B0 | A1 | | | | |
| 6 | | | | | | | | | | | | | | | | |
| 7 | | | | | | | | | - | T0 | - | T1 | S0 | S1w | T0 | T1 |
|  | | | | | | | | | | | | | - | H0 | - | H1 |

**Fig 26. Timing diagram for PWM operation.**

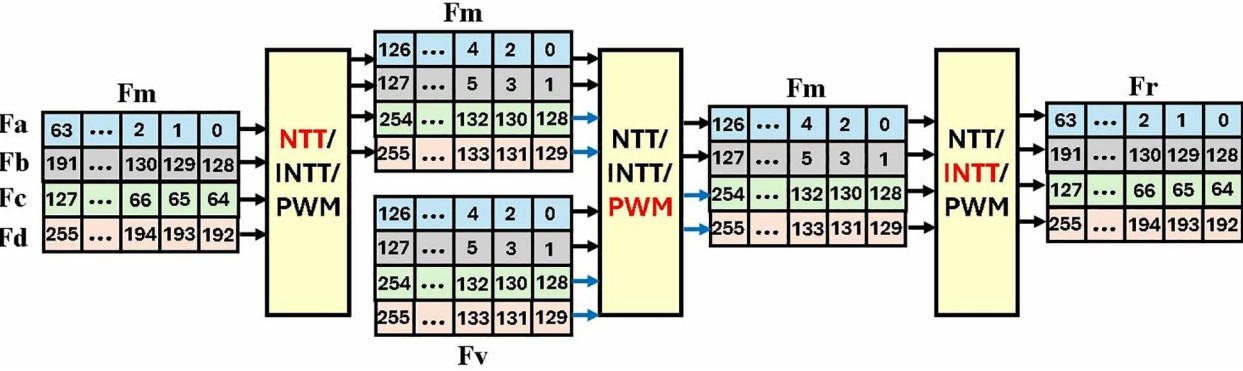

**Fig 27. Coefficient access scheme in MDC4NIP.**

## 4.6. TW management

Storage and management of pre-computed twiddle factors is crucial in NTT-based polynomial multiplications. As MDC2NIP has seven stages each comprising of one BU, instantiating seven BROMs results in large resource consumption. In this work, we have designed custom LUT-based distributed memories for all BUs.

The NTT, INTT and PWM computation require 384 TWs (128 + 128 + 128) in Kyber. In MDC2NIP, each butterfly unit has a different capacity of LUT-based distributed ROM memory (LRM) to store TWs for NTT, INTT and PWM computations. The first stage NTT operation requires 1 TW while the INTT operation requires 64 TWs. In PWM operation, TW is only needed in the second stage to compute $A_1.B_1.w$. Therefore, the second stage BU has a LRM storing 2 TWs for NTT, 32 TWs for INTT and 128 TWs for PWM. The remaining stages do not require TWs for PWM operation. Fig 28 presents the LRMs with breakdown of TWs for MDC2NIP architecture.

Our second proposed architecture MDC4NIP has seven stages with two BUs each. We have designed tailored LRMs according to the requirement of the specific BU. The memory in stage 1 of MDC2NIP is divided into two parts for the two BUs in stage 1 of MDC4NIP. In this way, all BUs operate independently and concurrently. Fig 29 presents the LRMs with breakdown of TWs for MDC2NIP architecture.

BU1 and BU2 require a common TW for NTT operation which is stored in both LRM1 and LRM2. 32 TWs are needed to perform INTT operation, while multiplication by w is not required in PWM computation in the first stage. In second stage, 2 TW for NTT and 32 TWs for INTT are required, with 1 TW for NTT and 16 TWs for INTT stored in LRM3 and LRM4, respectively. For PWM operation, BU3 only uses one constant TW "1" to be multiplied with input while BU4 performs the $A_1.B_1.w$ operation. Therefore, LRM3 and LRM4 store 1 and 128 precomputed values for PWM computation.

In 7 series FPGA, 6 input LUTS are used to store up to $2^6 = 64$ bits. This implies we can implement a 64x1 bit ROM using one LUT. Each stage in MDC2NTT and MDC4NTT has a different requirement of TWs and we have custom-built ROM in accordance to it. For example, the first stage in MDC2NTT, when working in NTT, INTT and PWM mode needs 1, 64,0 TWs. The total number of twiddle factors for this stage is 65 and the width of each coefficient is 12-bits, which are stored in a 65x12 bit ROM and uses 13 LUTS. The second stage requires 2, 32, 128 TW for NTT, INTT and PWM

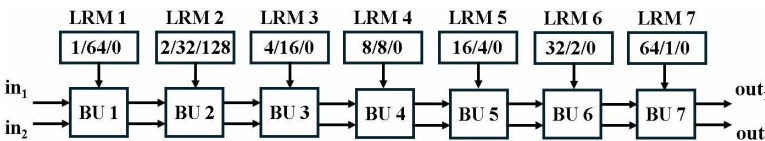

**Fig 28. TW management in MDC2NIP.**

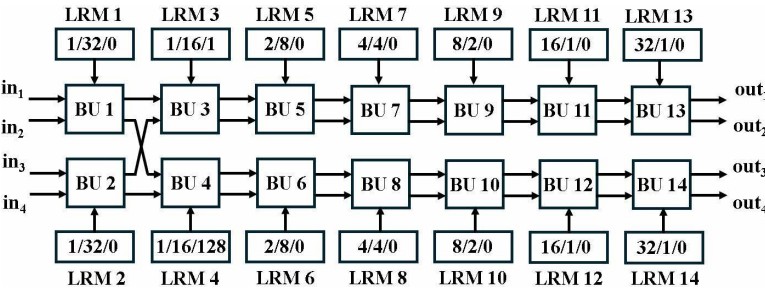

**Fig 29. TW management in MDC4NIP.**

operations. Total number of TWs is 162 and is implemented using 31 LUTs. The TWs for remaining stages occupy 4,3,4,6,13 respectively resulting in a total of 74 LUTs. The same number of LUTs are also required in MDC4NIP for TW storage, as it needs 384 TWs for NTT, INTT, PWM computation similar to MDC2NIP.

## 5. Result analysis

In this work we have developed two MDC-based NTT/INTT/PWM architectures, MDC2NIP and MDC4NIP for 512 security level of CRYSTALS-Kyber ($k=2$). The designs have been implemented on Artix-7 (XC7A100T-3) device using Verilog hardware description language. The Xilinx Vivado Design suite 2022.2 was used to synthesize, implement and perform functional verification and testing through the post-place and route simulations using Vivado simulator. A comprehensive testbench was developed to validate the functional correctness of the design's behavior over a series of clock cycles.

To provide a fair comparison, we have calculated the equivalent number of slices in each case using *ENS=(#LUTs/4)+(BRAMs x 196)+(DSPs x 100)* [19]. Area-Time product (ATP) is calculated as a metric for comparison which is the product of ENS and time for executing NTT/INTT/PWM operation.

Table 2 presents the resource comparison of our Karatsuba-Barrett based modular multiplier with different modular reduction architectures present in the literature. The techniques adopted to implement modular multiplier in this work, occupy the least number of resources when compared to various architectures in [15,19,45] and [46]. The implementation of LUT-based Karatsuba multiplier with shift-add based Barrett reduction, eliminates the use of DSP, minimizing the resource consumption. The lightweight operations reduce the design complexity while maintaining a high computational throughput.

The optimized MDC2NIP and MDC4NIP architectures are now compared with the works [17,29] and [22], which were discussed in the proposed architecture section for motivation. Comparison of resources between Radix -2 implementation in [29] with our Radix-2, 2-parallel and 4-parallel designs indicates a reduction of 73.7% and 52.4%, respectively. This indicates that the adopted optimization results in hardware efficiency.

When comparing our MDC2NIP to dual butterfly unit (DBU) in [17], the proposed design results in a 72% and 21.9% reduction in execution cycles of NTT/INTT and PWM operation. MDC4NIP results in 86% and 60.9% reduction in clock cycles when compared to DBU in [17], implying that our BRAM-free NTT core achieves considerably less execution time.

In comparison with 2 Bi-core in [22] to MDC2NIP, our proposed design results in an 80.3% reduction in ENS. MDC4NIP occupies 64.3% less resources than 2 Bi-core in [22]. This signifies that adopting Karatsuba multiplier with optimized Barrett reduction for modular multiplication results in a compact butterfly unit and ultimately in efficient NTT unit. Complete ATP comparison with State of the art architectures is given in the ensuing paragraphs.

Table 3 presents comparison of NTT/INTT/PWM architectures with former state of the art works in literature. Implementation of DSP and BRAM-free MDC2NIP design utilizes 1884 LUTs and 1076 FFs and achieves a maximum clock frequency of 200 MHz on targeted FPGA platform. Excluding pipeline stages, each NTT, INTT operation takes 128 clock cycles while PWM configuration utilizes 256 ccs for computation. Two parallel MDC chains, in MDC4NIP reduce the clock cycles for NTT, INTT and PWM operations to 64, 64 and 128 ccs respectively, at the expense of increase in resource usage. MDC4NIP occupies 3434 LUTs and 1845 FFs and operates at a maximum clock frequency of 184MHz.

**Table 2. Comparison of Modular Multiplier with different works.**

| Design | Techniques | LUT | FF | DSP | ENS |
|---|---|---|---|---|---|
| [45] | Karatsuba-Vedic multiplier, K-RED reduction | 228 | 174 | 0 | 77 |
| [15] | DSP-multiplier, K2-RED reduction | 62 | 42 | 1 | 180 |
| [19] | DSP multiplier, K2-RED reduction with and Look-up table | 50 | 34 | 1 | 150 |
| [46] | DSP-multiplier, Lookup table-based reduction | 82 | 0 | 1 | 120 |
| **[TW]** | **Karatsuba Multiplier, Barrett Reduction** | **147** | **36** | **0** | **40** |

**Table 3. Comparison of proposed NTT/INTT/PWM to state of the art designs.**

| Designs | Device | NTT/ INNT/PWM cycles | Freq (MHz) | Time (us) | #LUTs | #FFs | #DSPs | #BRAMs | ENS | AxT (ENS.ms) | TP (Mb/s) | TP/ slice (Kb/s) |
|---|---|---|---|---|---|---|---|---|---|---|---|---|
| [13] | Artix-7 | 512/576/256 | 161 | 3.2/3.6/1.59 | 1737 | 1167 | 2 | 3 | 1223 | 3.9/4.4/1.94 | 15 | 12.3 |
| (18)-a | Artix-7 | 940/1203/1289 | 115 | 8.17/10.5/11.2 | 360 | 145 | 3 | 2 | 879 | 7.2/9.2/9.8 | 2.9 | 3.3 |
| (18)-b | Artix-7 | 474/602/1289 | 115 | 4.12/5.2/11.2 | 737 | 290 | 6 | 4 | 1755 | 7.2/9.1/19.7 | 11.6 | 6.6 |
| (14)-a | Artix-7 | 904/904/647 | 190 | 4.8/4.8/3.4 | 948 | 325 | 1 | 2.5 | 827 | 3.97/3.97/2.81 | 5 | 6 |
| (14)-b | Artix-7 | 232/233/167 | 182 | 1.3/1.3/0.9 | 2543 | 792 | 4 | 9 | 2800 | 3.64/3.64/2.52 | 75.3 | 26.9 |
| (14)-c | Artix-7 | 69/71/47 | 172 | 0.4/0.4/0.3 | 9508 | 2684 | 16 | 35 | 10837 | 4.33/4.33/2.96 | 947.2 | 87.4 |
| (17)-a | Artix-7 | 906/906/649 | 350 | 2.6/2.6/1.85 | 407 | 411 | 0 | 7.5 | 1572 | 4.1/4.1/2.9 | 9.3 | 5.9 |
| (17)-b | Artix-7 | 458/458/328 | 342 | 1.3/1.3/0.96 | 590 | 671 | 0 | 7.5 | 1617.5 | 2.1/2.1/1.6 | 35.8 | 22.1 |
| (17)-c | Artix-7 | 234/235/169 | 334 | 0.7/0.7/0.5 | 1052 | 1058 | 0 | 13.5 | 2909 | 2/2/1.5 | 137 | 47.1 |
| [19] | Artix-7 | 456/456/265 | 300 | 1.5/1.5/0.88 | 1154 | 1031 | 2 | 0 | 645 | 0.98/0.98/0.57 | 31.6 | 48.9 |
| [27] | Artix-7 | 64/64/64 | 246 | 0.26/0.26/0.26 | 9211 | 9810 | 60 | 1 | 8948 | 2.3/2.3/2.3 | 184.5 | 20.6 |
| (22)-a | Artix-7 | 235/235/138 | 303 | 0.78/0.78/0.46 | 1170 | 1164 | 4 | 2 | 1208 | 0.94/0.94/0.56 | 61.9 | 51.2 |
| (22)-b | Artix-7 | 140/140/88 | 282 | 0.5/0.5/0.31 | 2283 | 2095 | 8 | 4 | 2388 | 1.19/1.19/0.74 | 193.4 | 80.9 |
| (22)-c | Artix-7 | 84/84/56 | 273 | 0.31/0.31/0.21 | 4619 | 4166 | 16 | 8 | 4809 | 1.5/1.5/1.0 | 624 | 129.8 |
| [23] | Artix-7 | 448/448/256 | 227 | 1.97/1.97/1.12 | 1005 | 599 | 2 | 0 | 519 | 1.0/1.0/0.6 | 24.3 | 46.8 |
| (47)-a | Artix-7 | 2016/2016/- | 307.5 | 6.55/6.55/- | 676 | 633 | 0 | 0 | 314 | 2.05/2.05/- | 7.3 | 23.2 |
| (47)-b | Artix-7 | 1009/1009/- | 306.4 | 3.28/3.28/- | 1137 | 1176 | 0 | 0 | 575 | 1.89/1.89/- | 14.6 | 25.4 |
| (47)-c | Artix-7 | 504/504/- | 304.7 | 1.65/1.65/- | 2081 | 2237 | 0 | 0 | 911 | 1.5/1.5/- | 29 | 31.8 |
| (48)-a | Artix-7 | – | – | – | 690 | 419 | 2.5 | 1 | 762.5 | – | – | – |
| (48)-b | Artix-7 | – | – | – | 1782 | 1027 | 9 | 4 | 2609.5 | – | – | – |
| (48)-c | Artix-7 | – | – | – | 6393 | 3604 | 33 | 16 | 9666.3 | – | – | – |
| **(TW)-a** | **Artix-7** | **128/128/256** | **200** | **0.64/0.64/1.28** | **1884** | **1076** | **0** | **0** | **471** | **0.3/0.3/0.6** | **37.5** | **79.6** |
| **(TW)-b** | **Artix-7** | **64/64/128** | **184** | **0.35/0.35/0.7** | **3410** | **1845** | **0** | **0** | **852.5** | **0.3/0.3/0.6** | **138** | **161.9** |

In [13] Xing et al. have proposed to perform NTT using two butterfly units. The unified butterfly units perform NTT, INTT and PWM operation by switching control code of muxes. (TW)-a and (TW)-b have less execution time and utilize considerably less resources when compared to [13]. ATP of (TW)-a and (TW)-b is significantly improved by 92.4%, 93.2% and 69% relative to [13] for NTT, INTT and PWM calculations, respectively.

Bisheh et al. in [18] have presented two NTT configurations represented here as [18]-a and [18]-b. The unified architecture is configured to perform NTT, INTT and PWM operations for polynomial multiplication. Our proposed designs (TW)-a, (TW)-b outperforms [18]-a with an improved ATP of 95.8% (NTT), 96.7% (INTT), 93.8% (PWM). ATP is improved up to 95.8% (NTT), 96.7% (INTT) and 96.9% (PWM) relative to [18]-b.

In [14], Yaman et al. have put forward three NTT architectures, lightweight (1 BU, [14]-a), balanced (4 BUs, [14]-b) and high-performance (16 BUs, [14]-c). The execution time when comparing (TW)-a, (TW)-b is greater than designs in [14] for certain cases. However, (TW)-a and (TW)-b surpass [14] by achieving lowest ATP. (TW)-a, (TW)-b has an improved ATP of 92.4% (NTT/INTT), 78.6% (PWM) when compared to [14]-a, 91.8% (NTT/INTT), 76% (PWM) in comparison to [14]-b and 93% (NTT/INTT), 79% (PWM) relative to [14]-c.

Three NTT configurations with single butterfly unit [17]-a, one dual butterfly unit [17]-b and two dual butterfly unit [17]-c, are presented by Alhassani et al. Interpretation of the results reveal that our design has an increased execution time for PWM operation when comparing [17]-b with (TW)-b and [17]-c with (TW)-a, TW-c. The designs in [17] heavily rely on BRAMS for storing coefficients. This increases ENS considerably and ultimately results in high value of ATP. (TW)-a,

(TW)-b exhibits improved ATP by 92.7% (NTT/INTT), 79.3% (PWM) when compared to [17]-a, 85.8% (NTT/INTT), 62.5% (PWM) relative to [17]-b and 85% (NTT/INTT), 60% (PWM) when compared to [17]-c.

The design in [19] has an improved ATP for PWM, when compared to (TW)-a and (TW)-b by 5%. However, the ATP of proposed NTT/INTT operation outshines that of [19], with 69.4% (NTT/INTT) improvement in comparison with (TW)-a and (TW)-b. Since NTT, INTT operations show significant increase and the PWM computation demonstrates marginal decrease in performance, we can conclude that our designs are more efficient than [19].

Duong et al. [27] have proposed a Multipath Delay Feedback (MDF) architecture for different parameter sets (n,q). The design utilizes considerable DSPs which results in large resource utilization. Comparison between (TW)-a, (TW)-b with [27] depicts an increase in ATP performance by 87% (NTT/INTT), 74% (PWM).

In [22] Li et al. have proposed three configurable based NTT designs, which can perform NTT/INTT and PWM operations. (TW)-a and (TW)-b attain an improved ATP by 74.8% (NTT/INTT), 18.9% (PWM) when compared to [22]-b. Our designs outperform [22]-c by achieving better ATP by 80% (NTT/INTT), 40% (PWM). Result comparison shows that performance is improved by 68% for NTT/INTT operations when [22]-a is compared with (TW)-a and (TW)-b. However, [22]-a have 6% improved ATP for PWM operation in comparison with (TW)-a and (TW)-b. Despite this minimal decrease, proposed design stands out due to performance achievements in NTT/INTT operation.

A BRAM-free iterative architecture NMI-NTT has been proposed in [23]. (TW)-a and (TW)-b outperforms [23] by achieving a 70% better ATP in NTT/INTT operation, while achieving a comparable ATP for PWM operation.

In [47], the authors proposed a unified Butterfly Unit (UBU) that combines interleaved multiplication, radix-4 processing, and resource-sharing techniques to perform NTT and inverse NTT (INTT) operations efficiently. The work implements three NTT/INTT architectures lightweight [47]-a, balanced [47]-b and low latency [47]-c designs, each consisting of $2^2$, $2^3$ and $2^4$ butterfly units. In comparison with [47]-a, [47]-b and [47]-c, (TW)-a and (TW)-b achieves a 85.3%, 84% and 80% improved ATP for NTT/INTT operation, respectively. As, PWM operation is not implemented in [47], comparison of ATP for PWM has not been provided.

A novel design for modular reduction is presented in [48]. The compact design is used to implement a butterfly unit for CRYSTALS-Kyber. To evaluate the efficiency of the proposed design, the authors in [48] replaced the butterfly unit in an existing open-source NTT-based polynomial multiplication [14] for Kyber, with their own optimized butterfly unit. This allowed for a direct comparison of area utilization, demonstrating a significant reduction in LUT usage. While preliminary findings indicate that the latency of the design in [48] is lower than that of the original open-source implementation, the clock frequency and number of clock cycles were not reported in [48]. Therefore, the comparison of our proposed MDC2NIP and MDC4NIP with [48] is limited to area resource utilization only. Proposed (TW)-a when compared to [48]-a, [48]-b and [48]-c, achieves an area reduction of 51.7%, 81.9% and 95.12%, respectively. A reduction of 67.3%, 91.8% in resources is obtained when comparing (TW)-b with [48]-a and [48]-b.

Proposed architectures MDC2NIP and MDC4NIP achieve lowest ATP for NTT/INTT designs till date when compared to the prior works. PWM operation achieves comparable performance, provided no additional hardware is used for its implementation. This is primarily credited to designing a shared architecture for performing NTT, INTT and PWM operations. Our approach of implementing a FIFO-based memory and LUT-based multiplications has resulted in a low ENS value of architectures. Results analysis reveals that MDC2NIP has an increased execution time but utilizes half of its resources when compared to MDC4NIP. However, both architects achieve similar ATP, and both can be employed to provide a tradeoff between area and time.

Table 3 also presents throughput (TP) and throughput per slice (TP/slice) for NTT operation. TP and TP/A are calculated using {(clock frequency x no. of bits)/latency} and (throughput/ENS), respectively. The proposed architecture and the others listed in the comparison table have reported different cycles for NTT, INTT and PWM operation. As the number of execution cycles are different the throughput for each operation will also be different. TP and TP/slice for INTT and PWM operation can be calculated using the formula given. It is important to note that TP and TP/slice are not reported in works

**Table 4. Resource utilization of modules in proposed implementations.**

| Modules | MDC2NIP | | MDC4NIP | |
|---|---|---|---|---|
| | LUTs | FIFOs | LUTs | FIFOs |
| Butterfly[2] | 1302 | 420 | 2604 | 840 |
| Mod add[1] | 23 | 0 | 23 | 0 |
| Mod sub[1] | 10 | 0 | 10 | 0 |
| Int Mul[1] | 84 | 0 | 84 | 0 |
| Mod red[1] | 63 | 36 | 63 | 36 |
| Commutator[2] | 125 | 54 | 250 | 110 |
| TWROM[2] | 74 | 28 | 74 | 28 |
| FIFO (MDC)[2] | 95 | 284 | 193 | 577 |
| FIFO (Memory)[2] | 288 | 290 | 288 | 290 |
| Total | 1884 | 1076 | 3409 | 1845 |

1. Resources of one unit.

2. Resources of all units present in architecture.

listed in the comparison table and have been calculated based on the reported clock frequency, latency (clock cycles) and the number of bits as input per cycle in the respective designs.

The results of TP/slice indicate that the proposed MDC4NIP outperforms the state of the art architectures and demonstrates superior efficiency in achieving the desired performance with fewer hardware resources. MDC4NIP has an increased TP/A by 92.4%, 97.9%, 95.9%, 96.3%, 83.4%, 46%, 96.3%, 86.3%, 70.9%, 69.8%, 87.3%, 68.4%, 50%, 19.8%, 71.1%, 85.7%, 84.3%, 80.3% compared to [13,18]-a, [18]-b, [14]-a, [14]-b, [14]-c, [17]-a, [17]-b, [17]-c, [19,22,27]-a, [22]-b, [22]-c, [23,47]-a, [47]-b, [47]-c, respectively. MDC2NIP outperforms [13,18]-(a), [18]-(b), [14]-(a), [14]-(b), [17]-a, [17]-b, [17]-c, [19,22,27]-a, [23,47]-a, [47]-b, [47]-c by 84.5%, 95.8%, 91.7%, 92.4%, 66.1%, 92.8%, 72.2%, 40.8%, 38.5%, 74.8%, 35.7%, 41.2%, 70.9%, 68.1%, 60%, respectively.

Proposed MDC2NIP consists of seven butterfly units (BUs). Table 4 presents the resources of all the butterfly units in MDC2NIP that is 1302 LUTs and 420 FFs. MDC4NIP consists of fourteen butterfly units and hence the BUs occupy twice the resources than in MDC2NIP, that is 2604 LUTs and 840 FFs.

In MDC2NIP design, 7 BUs and 6 commutators occupy 1302 and 125 LUTs. Distributed LUT-based BROM for twiddle factor storage (TWROM), FIFO in MDC chain [FIFO(MDC)] and FIFO for storing input/output coefficients of NTT, INTT and PWM operation [FIFO(Memory)] consumes 74, 95 and 288 LUTs. This equals a total of 1884 LUTs. In MDC4NIP architecture, 14 BUs, 12 commutators, TWROM, FIFO (MDC) and FIFO (Memory) occupy 2604, 250, 74, 193 and 288 LUTs, which results in a total of 3409 LUTs.

Our implemented MDC2NIP and MDC4NIP can be deployed in IOT constrained devices due to its lightweight, compact architecture and low power consumption. The two designs use only 2.97% and 5.4% resources on the selected hardware platform Artix-7 comprising of 63400 LUTs and 126800FFs. Additionally, MDC2NIP consumes total on-chip power of 843mW, while the power consumption of MDC4NIP is 1194mW, making them power efficient designs.

## 6. Conclusion

CRYSTALS-Kyber, a lattice-based cryptography scheme, has been standardized by NIST as a public-key encryption and key-establishment algorithm in 2022. This research focusses on accelerating the NTT-based polynomial multiplication in Kyber. Two pipelined multi-path delay commutator (MDC) architectures have been implemented, with two (MDC2NIP) and four (MDC4NIP) parallel paths. The MDC4NIP implementation occupies twice the resources of MDC2NIP but uses half the time. Both architectures achieve a similar area-time product, allowing for a trade-off between area and time. Proposed

architectures achieve the lowest ATP for NTT/INTT operations by 68% when compared to former works. A near-equivalent ATP is achieved for PWM computations while using the same resources as NTT/INTT computation. MDC2NIP uses up to 29% reduced hardware resources than the state of the art designs.

## 7. Future work

Number theoretic transform (NTT) is employed by CRYSTALS-Kyber for computing polynomial multiplication. In this paper we have implemented two resource-efficient and high performance NTT architectures. Our pipelined designs, MDC2NIP and MDC4NIP, outperform the state of the art architectures, as discussed in section 5. In future a unified butterfly architecture will be designed, supporting both Kyber and Dilithium. This will enable our proposed architectures to function efficiently in a multi-hybrid cryptographic environment, enhancing scalability. Future research will incorporate power consumption analysis and evaluate the resilience of our proposed architecture against side-channel attacks. The implemented cores MDC2NIP and MDC4NIP will also be deployed to implement CRYSTALS-Kyber algorithm, by optimizing the bottleneck operations such as key-generation, encapsulation and decapsulation.

## Author contributions

**Conceptualization:** Ayesha Waris.

**Data curation:** Ayesha Waris.

**Formal analysis:** Ayesha Waris.

**Investigation:** Ayesha Waris.

**Methodology:** Ayesha Waris.

**Project administration:** Arshad Aziz.

**Resources:** Ayesha Waris, Arshad Aziz.

**Software:** Ayesha Waris.

**Supervision:** Arshad Aziz, Bilal Muhammad Khan.

**Validation:** Ayesha Waris.

**Visualization:** Ayesha Waris.

**Writing – original draft:** Ayesha Waris.

**Writing – review & editing:** Arshad Aziz, Bilal Muhammad Khan.

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
