## [Decision Letter · Decision Letter 0]

19 Feb 2025

PONE-D-24-46384Area-Time Efficient, Parallel-Pipelined Number Theoretic Transform for CRYSTALs-KyberPLOS ONE

Dear Dr. Waris,

Thank you for submitting your manuscript to PLOS ONE. After careful consideration, we feel that it has merit but does not fully meet PLOS ONE’s publication criteria as it currently stands. Therefore, we invite you to submit a revised version of the manuscript that addresses the points raised during the review process.

**Please address all the comments and issues raised by the reviewers and ** submit your revised manuscript by Apr 05 2025 11:59PM. If you will need more time than this to complete your revisions, please reply to this message or contact the journal office at plosone@plos.org . Please include the following items when submitting your revised manuscript:

We look forward to receiving your revised manuscript.

Kind regards,

Muhammad E. S. Elrabaa

Academic Editor

PLOS ONE

Journal Requirements:

Additional Editor Comments:

Please prepare a major revision of the manuscript and address all the comments of the reviewers.

Reviewers' comments:

Reviewer's Responses to Questions

**Comments to the Author**

1. Is the manuscript technically sound, and do the data support the conclusions?

Reviewer #1: Yes

Reviewer #2: Partly

2. Has the statistical analysis been performed appropriately and rigorously? 

Reviewer #1: Yes

Reviewer #2: Yes

3. Have the authors made all data underlying the findings in their manuscript fully available?

Reviewer #1: No

Reviewer #2: No

4. Is the manuscript presented in an intelligible fashion and written in standard English?

Reviewer #1: Yes

Reviewer #2: Yes

5. Review Comments to the Author

Reviewer #1: This article demonstrate an area-time efficient implementation of Kyber on FPGA device. It is a timely contribution and seems to be better than SOTA implementation. There are some minor comments to be addressed before it can be accepted for publication.

1) The idea of MDC is not new, it was already used by Ref. 23 and this paper (10.1109/TC.2023.3296899). What is the novelty of the proposed MDC (Section 4.3 and 4.4) compared to these existing works?

2) Section 4.1 is presenting many known results, they are not new and should not be considered as contributions/proposed ideas.

3) The article is not written in a way easy to understand the contributions. The authors tend to report the detailed implementation techniques without making comparison with the SOTA. This is clearly shown in the contribution paragraph, the authors only mentioned some implementations that they have performed, without discussing why these are considered novel. Note that although the authors can achieve better area-time efficiency, it is also important to discuss why this happens.

4) It is advisable to share the source code so allow reproducible research.

Reviewer #2: This article presents an area-time efficient, parallel pipelined NTT for CRYSTALS-KYber. It presents two parallel architectures based on the MDC approach and resource sharing technique. Several optimization techniques have been incorporated in low-level multiplier architecture and twiddle factor storage. It is synthesized on Artix-7 FPGA platforms. It is a DSP and BRAM-free design delivered up to 68% improvement in are-time product. However, the following concerns need to be addressed:

1.The title needs to be revised; usually, it does not contain a comma, so it is better to remove it after ‘efficient’. The word ‘parallel-pipelined’ does not make sense; it is better to use only pipelined, change ‘CRYSTALs’ to CRYSTALS.

2.The abstract must include details of the implementation platform, such as the FPGA family/device and the associated tools.

3.How the functional verification of the design has been conducted, add details in the implementation section.

4.How the lookup table-based multiplier is efficient than the strategies adopted in Montgomery and interleaved multipliers presented in (1) point multiplication accelerator for arbitrary Montgomery curves, (2) efficient reconfigurable modular multipliers for post-quantum digital signatures, and (3) efficient soft core multipliers for digital signatures?

5.It is demonstrated in Table 3 that the proposed butterfly with FIFO consumes around 1302, 420 LUTs in MDC2NIP, and even it jumps to 2604, 840 in case of MDC4NIP. How the overall design can consume just 1884 LUTs. Clarify how many BU you are using in the MDC2 and MDC4. Add details of your BU, such as LUTs, FFs, and clock cycle consumption. Moreover, how are you handling div by 2 in GSBU?

6.As the design is BRAM free, how many twiddle factors do you need to store in NTT and INTT, and how many LUTs/FFs are being consumed for just this storage?

7.In Kyber, how many PWM are required, and in your setting, how many clock cycles are consumed in its computation?

8.Some recent work must be added and compared in Table 2, such as (1) A better Kyber Butterfly for FPGAs, (2) Efficient Number Theoretic Transform Architecture for CRYSTALS-Kyber, etc.

9.Most of the schemes are now hybrid, so how will your design scale in a multi-scheme environment where lattice and hash-based schemes are integrated?

10.Add throughput per slice figure in Table 2 and compare your design with the listed designs on this factor as well.

11.Power consumption is very important, add power consumption stats and compare it with the listed designs.

12.Side-channel attacks are very important nowadays. How robust is your design against these attacks (power and timing)?

6. PLOS authors have the option to publish the peer review history of their article (what does this mean? ). If published, this will include your full peer review and any attached files.

**Do you want your identity to be public for this peer review?** For information about this choice, including consent withdrawal, please see our Privacy Policy .

Reviewer #1: No

Reviewer #2: No

---

## [Author Response · Author response to Decision Letter 1]

6 Mar 2025

Reviewer 1 and Reviewer 2: The Authors appreciate the feedback of the Reviewers and have comprehensively addressed the comments in the revised manuscript.

---

## [Decision Letter · Decision Letter 1]

24 Mar 2025

PONE-D-24-46384R1Area-Time Efficient Pipelined Number Theoretic Transform for CRYSTALS-KyberPLOS ONE

Dear Dr. Waris,

Thank you for submitting your manuscript to PLOS ONE. After careful consideration, we feel that it has merit but does not fully meet PLOS ONE’s publication criteria as it currently stands. Therefore, we invite you to submit a revised version of the manuscript that addresses the points raised during the review process.

We look forward to receiving your revised manuscript.

Kind regards,

Muhammad E. S. Elrabaa

Academic Editor

PLOS ONE

**Journal Requirements:**

**Additional Editor Comments:**

Please address the remaining points from the previous review **in the manuscript,** not just as rebuttals.

Also, consider sharing, fully or partially, your codes to encourage reproducible research in the community.

Reviewers' comments:

Reviewer's Responses to Questions

**Comments to the Author**

1. If the authors have adequately addressed your comments raised in a previous round of review and you feel that this manuscript is now acceptable for publication, you may indicate that here to bypass the “Comments to the Author” section, enter your conflict of interest statement in the “Confidential to Editor” section, and submit your "Accept" recommendation.

Reviewer #1: All comments have been addressed

Reviewer #2: (No Response)

2. Is the manuscript technically sound, and do the data support the conclusions?

Reviewer #1: Yes

Reviewer #2: Yes

3. Has the statistical analysis been performed appropriately and rigorously? 

Reviewer #1: Yes

Reviewer #2: Yes

4. Have the authors made all data underlying the findings in their manuscript fully available?

Reviewer #1: No

Reviewer #2: No

5. Is the manuscript presented in an intelligible fashion and written in standard English?

Reviewer #1: Yes

Reviewer #2: Yes

6. Review Comments to the Author

**Reviewer #1: ** The authors had addressed all my comments. It is ready for publication in current form. It is advisable that the authors can share fully or partially their codes to encourage reproducible research in the community.

**Reviewer #2: ** The author(s) partially addressed the previous concerns, these are:

1. The authors provided details in the response letter for the concern " How the lookup table-based multiplier

is efficient than the strategies adopted in

Montgomery and interleaved multipliers

presented in (1) point multiplication

accelerator for arbitrary Montgomery

curves, (2) efficient reconfigurable

modular multipliers for post-quantum

digital signatures, and (3) efficient soft

core multipliers for digital signatures?". But this discussion is not reflected in any part of the revised manuscript. It is important to add this because this will add another dimension that reduction can be done using interleaved fashion.

2. Response against another concern " Some recent work must be added and

compared in Table 2, such as (1) A better

Kyber Butterfly for FPGAs, (2) Efficient

Number Theoretic Transform

Architecture for CRYSTALS-Kyber, etc." is not satisfactory. The better Kyber paper also reported a complete NTT/INTT implementation results. Both these manuscript are recent proposals so must be compared.

3. Power consumption figures are missing again. Vivado provides a simple way to measure the power so it is better to add power consumption figures.

7. PLOS authors have the option to publish the peer review history of their article (what does this mean? ). If published, this will include your full peer review and any attached files.

**Do you want your identity to be public for this peer review?** For information about this choice, including consent withdrawal, please see our Privacy Policy .

Reviewer #1: No

Reviewer #2: No

---

## [Author Response · Author response to Decision Letter 2]

29 Mar 2025

Reviewer 2: The Authors appreciate the feedback of the Reviewer and have comprehensively addressed the comments in the revised manuscript.

---

## [Editor Report · Decision Letter 2]

6 Apr 2025

Area-Time Efficient Pipelined Number Theoretic Transform for CRYSTALS-Kyber

PONE-D-24-46384R2

Dear Dr. Waris,

We’re pleased to inform you that your manuscript has been judged scientifically suitable for publication and will be formally accepted for publication once it meets all outstanding technical requirements.

Kind regards,

Muhammad E. S. Elrabaa

Academic Editor

PLOS ONE

Additional Editor Comments (optional):

Please provide the code along the manuuscript.
---

## [Editor Report · Acceptance letter]

PONE-D-24-46384R2

PLOS ONE

Dear Dr. Waris,

I'm pleased to inform you that your manuscript has been deemed suitable for publication in PLOS ONE. Congratulations! Your manuscript is now being handed over to our production team.

Kind regards,

on behalf of

Dr. Muhammad E. S. Elrabaa

Academic Editor

PLOS ONE